# Literati Ingredients in the 17th-Century Chinese Christian Paintings

Jie Xiao

School of Art, Southeast University, Nanjing 211189, China; qiuyeweiyu@163.com

**Abstract:** In this paper, the modification methods of the Chinese Christian painting created by the missionaries in the late Ming Dynasty (1573–1644) were analyzed with the Chinese Catholic studies of the "*Song nianzhu guicheng*" and the "*Tianzhu Jiangsheng Chuxiang Jingjie*". After carefully studying the differences between the Chinese Christian painting and the original European version, the study shows that these Chinese Christian paintings were integrated with the Chinese literati paintings' elements and literati symbols, which include the "*Yudiancun*" (raindrop texture stroke), "*Pimacun*" (hemp-fiber texture stroke), "landscape screen" (painted screens with natural landscapes), and the mark of Chinese famous literati such as Dong Qichang. These adjustments conducted by missionaries aimed to make religious paintings more in line with literati aesthetics, which could build connections between the missionaries and the literati community for proselytization. However, the missionaries neglected that the literati community certainly would not sacrifice the existing social order and the vested interest brought by the current Confucian culture to support new ideas of "liberty" and "equality" in the Catholic doctrine, which caused a huge setback in the missionary work since the Nanjing Teaching Case in 1616. This research makes significant contributions to the understanding of cultural exchanges in the 17th century through a detailed exploration of the adjustments made by missionaries in the visual representations within Chinese Catholic literature.

**Keywords:** Chinese Christian painting; literati painting elements; literati symbol; missionary work in the late Ming Dynasty

## 1. Introduction

In the 17th century, a group of European missionaries, including Catholics and Protestants, such as Matteo Ricci (an Italian Jesuit priest), Jean de Rocha (a Portuguese Jesuit), and Giulio Aleni (an Italian Jesuit missionary and scholar), with high cultural literacy, traveled to China for missionary work. Actively engaging with Chinese culture during the missionary process, they produced two religious books infused with Chinese aesthetics. These include "*Song nianzhu guicheng* 誦念珠規程 *(Rules for Reciting the Rosary)*,"[1] published in Nanjing in 1619. This publication, based on a series of European engravings, served to illustrate a Chinese translation of the Rosary by the Portuguese Jesuit Jean de Rocha. Another study, "*Tianzhu Jiangsheng Chuxiang Jingjie* 天主降生出像經解 *(Explanations of the Scripture with Images of the Lord of Heaven Incarnate)*," published in Jinjiang (Quanzhou) in 1637, comprises 56 engravings and a map of Jerusalem. This publication also draws inspiration from European prints found in Nadal's study[2]. In the illustrations of these two books, we can observe that missionaries employed traditional composition and techniques from Chinese painting, offering a reinterpretation of Christian paintings. This adaptation goes beyond the mere acquisition of painting techniques. It reflects a conscious effort by missionaries to integrate literati elements (e.g., landscape) into Christian paintings, demonstrating their attempt to incorporate religious paintings into the literati painting system that emphasizes personal expression.

Currently, scholars from both China and the West have extensively researched the developmental process of Chinese Christian paintings, primarily focusing on the analysis of

the purposes, processes, and influences of these paintings. Gianni Criveller introduced the Chinese elements in the 17th-century publications "*Song nianzhu guicheng*" and "*Tianzhu Jiangsheng Chuxiang Jingjie*", advocating that missionaries used these improvements to attract the curiosity of the emperor and the literati class to induce them to join Catholicism. (Criveller 2010). Michael Sullivan and James Cahill discussed the impact of Western Jesus paintings on Chinese art and analyzed the learning and application of Western painting by Chinese painters such as You Wenhui (1575–1630) and Chang Hung (1577–1668) (Cahill 1982, p. 56) during the Ming Dynasty (Sullivan 2014, pp. 58–77). Chu Xiaobai explored the changes in the image of Jesus from the late Ming to the end of the Qing Dynasty and analyzed the shift in the attitude of the Chinese towards Catholicism behind these changes (Chu 2001, pp. 35–52).

Although these studies illustrated the changes in Christian paintings and the incorporation of some Chinese elements in the late Ming Dynasty, they did not further investigate the sources of these Chinese elements being incorporated. Therefore, this paper uses image analysis to examine Christian paintings and Chinese paintings comparatively and finds that the missionaries' improvement of the Christian paintings mainly came from literati painting elements and literati symbols, which reflected their urgent psychological desire to establish connections with Chinese literati, who were influential educators of scholars and officials during that period.

## 2. The Application of the Literati Painting Technique in Christian Paintings

Jesus, as the central figure in Catholicism, has developed a fixed pattern over centuries in the Western context. In religious paintings, the image of Jesus is typically positioned prominently at the center of the composition, and the artists usually dedicate considerable effort to depicting the facial features and posture realistically. However, due to significant cultural differences between China and Europe, imposing Christian paintings unrelated to Chinese cultural attributes on the Chinese people appears impractical. Therefore, early missionaries sought to transform Christian paintings by incorporating elements of Chinese culture.

Early Christian missionaries, including Matteo Ricci (1552–1610), recognized that written communication was more effective than oral communication in disseminating the fundamental beliefs of Christianity to the Chinese people, due to the Chinese people's enthusiasm for reading new content and the expressive power of Chinese ideograms. Therefore, many missionaries began to translate and publish Catholic books after the encouragement of Matteo Ricci. Another missionary named Michel Ruggier (1543–1607) was responsible for translating and publishing the *Lord's Prayer*, *Hail Mary*, and *Ten Commandments* into Chinese. In addition to these translations, other missionaries such as Jean de Rocha (1566–1623), Giulio Aleni (1582–1649), and Johann Adam Schall von Bell (1591–1666) also published religious studies with illustrations that incorporated Chinese artistic motifs. Through the analysis of these illustrations, it is evident that the missionaries did not simply adopt the artistic painting techniques of Chinese painting when modifying the Christian paintings. Instead, they used common artistic painting techniques and symbols found in literati painting to reshape these Christian paintings.

### 2.1. Yudiancun 雨点皴 (Raindrop Texture Stroke)

In the practice of depicting natural landscapes, Chinese literati painters formed a unique language of texture patterns composed of brushstrokes by summarizing and abstracting the different geological features, which was referred to as "*Cunfa* 皴法". The different types of *Cunfa* emerged with variations in topography involving using a brush to create short and pointed strokes that resemble raindrops, aiming to portray the grandeur and solidity of rocks. For instance, in the painting "*Traveling through Streams and Mountains Painting* 溪山行旅圖" by Fan Kuan (950–1032), he outlined the contours of the rocks with bold short lines at first and used a continuous series of "*Yudiancun*" to present the weightiness of the landscape (Figure 1).

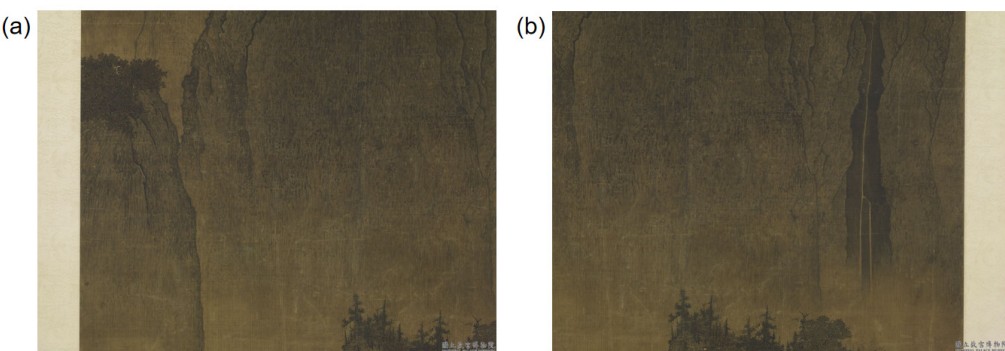

**Figure 1.** Yudiancun in Chinese paintings. Fan Kuan, (**a**,**b**) are the part of *Traveling through Streams and Mountains Painting* (950–1032), 206.3 × 103.3 cm. Courtesy of the National palace museum, Taibei.

This method of painting was also found in religious books published by missionaries in the late Ming Dynasty. "*Song nianzhu guicheng*", published by missionary Jean de Rocha, contains 15 illustrations, each corresponding to "*The Fifteen Mysteries of the Catholic Jesus*". The prevailing belief is that these illustrations in the "*Song nianzhu guicheng*" are based on "*Evanglicae Historiae Imagines*"[3] and correspond pairwise with it. However, Zhu Lihan's examination revealed that the "*Four Ends of Glory and Blessing*" in the "*Song nianzhu guicheng*" correspond to two illustrations in the "*Evanglicae Historiae Imagines*", while the "*Fifth End of Glory*" may have been inspired by Albrecht Dürer's (1471–1528) woodcut from 1510, titled "*The Assumption and Coronation of the Virgin*" (Zhu 2022, p. 29). But, after conducting the careful reexamination and comparison of the illustrations in the "*Song nianzhu guicheng*" with the "*Evanglicae Historiae Imagines*" and other European paintings, we found that the illustrations in the "*Song nianzhu guicheng*" not only abandon certain European painting techniques, such as chiaroscuro[4], but also incorporate the artistic painting technique of traditional Chinese woodblock prints and introduce some common painting techniques from literati painting.

The "*Christ praying in the garden of Gethsemane*" from the "*Song nianzhu guicheng*" was taken as an example to analyze the application of "*Cunfa*". First, in "*Song nianzhu guicheng*", the number of characters is relatively small and concise, and the composition mainly features mountains, rocks, trees, and auspicious clouds. Compared to European paintings, the proportion of space between Jesus and the background in "*Song nianzhu guicheng*" is rather peculiar, which makes Jesus appear smaller than the rocks in the foreground. This spatial treatment is entirely based on the "*Retreat Painting* 隱逸圖" in Chinese literati painting, where figures are placed in a vast and detailed landscape. The "*Chanting to the Moon Under Pines* 松間吟月" used the perspective technique of "foreground, midground, and background", commonly seen in Chinese landscape painting, to depict spatial depth (Figure 2). Second, the rocks in "*Song nianzhu guicheng*" also employ the painting language of "*Yudiancun*". The rocks in "*Song nianzhu guicheng*" are depicted using brushstrokes of inverse edges and central edges in the foreground, with vertical short lines drawn to represent various facets of the rocks. This technique is similar to "*Traveling through Streams and Mountains Painting*" composed by Fan Kuan and "*Spring Mountains and Auspicious Pines* 春山瑞松圖" composed by Mi Fu (1051–1107), which was widely known as "*Yudiancun*" (Figure 3). Moreover, the Tang Dai (1673–1752) and Dong Qichang (1555–1636) also confirmed that the "*Yudiancun*". is a common painting language of literati painting in the books "*Huishifawei* 繪事發微" (Dai 1987, p. 24) and "*Huachanshisuibi* 畫禪室隨筆" (Dong 1999, p. 124), which makes the "*Yudiancun*" used in the "*Christ praying in the garden of Gethsemane*" from the "*Song nianzhu guicheng*" evidence of utilizing the artistic painting technique of literati painting by the missionary.

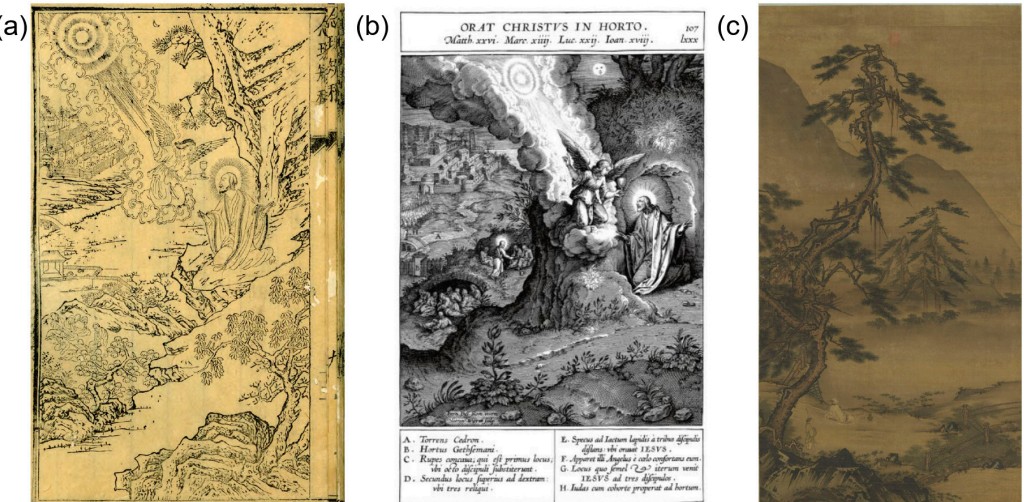

**Figure 2.** Comparison between "*Christ praying in the garden of Gethsemane*" and Scholar's Hermit Images. (**a**) *Christ praying in the garden of Gethsemane* (drawn from Rocha 1619, p. 40). (**b**) *Christ praying in the garden of Gethsemane* (drawn from Nadal 1593, p. 116). (**c**) Ma Yuan, *Chanting to the Moon Under Pines* (1190–1222), 144.6 × 76.3 cm. Courtesy of the National palace museum, Taibei.

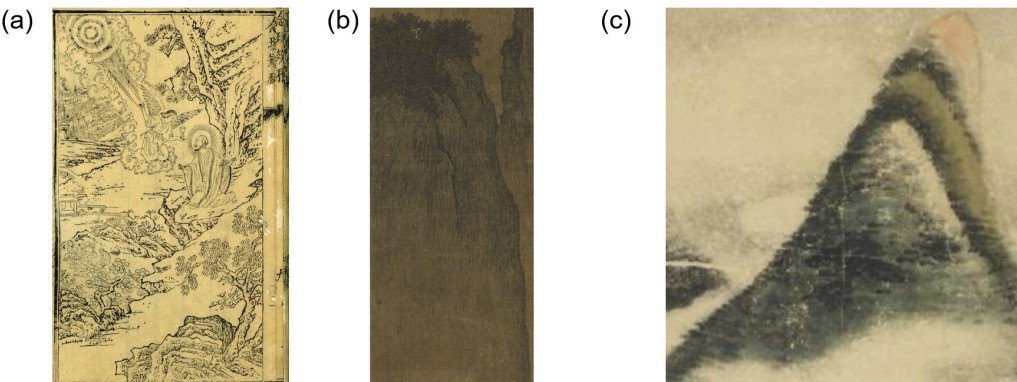

**Figure 3.** Comparison between the "*Yudian Cun*" in "*Christ praying in the garden of Gethsemane*" and Scholar Paintings. (**a**) The rock of the *Christ praying in the garden of Gethsemane* (drawn from Rocha 1619, p. 40). (**b**) Fan Kuan, The rock of *Traveling through Streams and Mountains Painting* (960–1127). 206.3 × 103.3 cm. Courtesy of the National palace museum, Taibei. (**c**) Mi Fu, The rock of *Spring Mountains and Auspicious Pines* (1051–1107), 35 × 44.1 cm. Courtesy of the National palace museum, Taibei.

## 2.2. *Pimacun* 披麻皴 *(Hemp-Fiber Texture Stroke)*

Besides the Chinese literati painting technique of the "*Yudiancun*", the "*Pimacun*" also employed to depict the scenes of Christian paintings in the "*Song nianzhu guicheng*" illustrations. "*Pimacun*" is a common artistic language in Chinese landscape painting, which is composed of "*Chang pimacun* 長披麻皴" with long lines and "*Duan pimacun* 短披麻皴". with short lines. The "*Yudiancun*" was often used to depict the basic appearance of the southern Chinese mountains, and Dong Yuan 董源, from the 9th century, is a representative figure of the "*Duan pimacun*" technique. In his painting "*Residents on the Outskirts of the Capital* 龍宿郊民圖", the "*Duan pimacun* 短披麻皴" is widely used to depict the light and shadow and the three-dimensionality of the rocks, which aptly portrays the natural scenery of southern China. Besides that, Huang Gongwang (1269–1354) is a representative figure of the "*Chang pimacun*" technique. His painting "*Dwelling in the Fu-chun Mountains* 富春山居圖" depicts the beautiful scenery around the Fuchun River in Zhejiang in a long scroll format. The "*Chang pimacun* technique" is arranged from top to bottom along the

direction of the mountains, and the variation in the density of the lines and the intensity of the ink felicitously portrays the outline of these rocks (Figure 4).

(a) 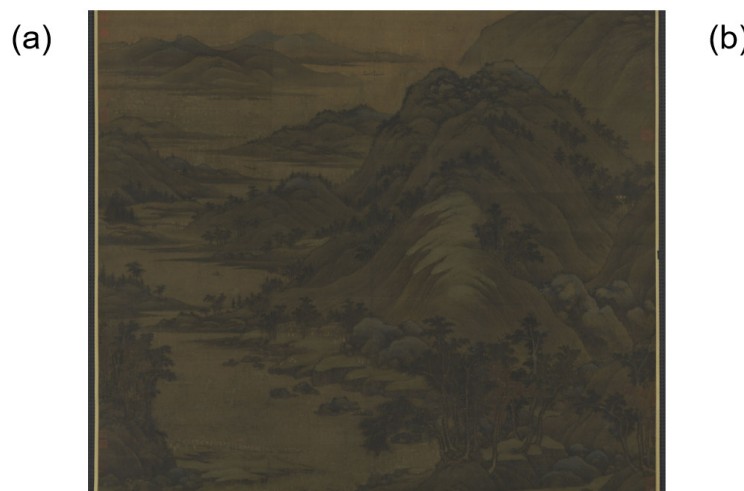 (b) 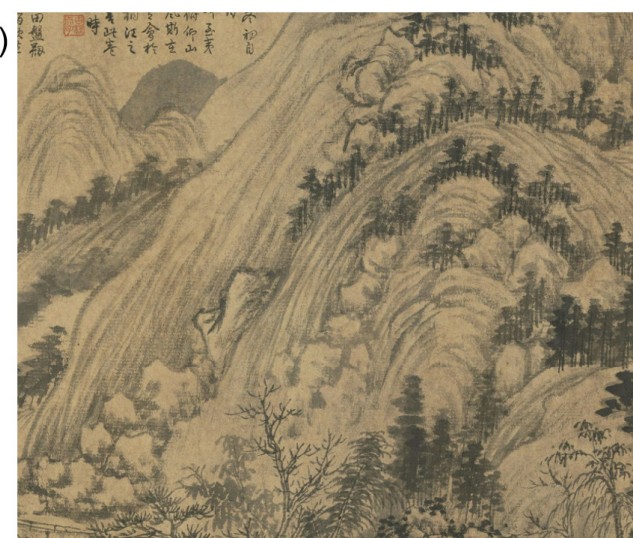

**Figure 4.** *Pimacun* in Chinses paintings. (**a**) Dong Yuan, The part of *Residents on the Outskirts of the Capital* (about 934–962), 156 × 160 cm. Courtesy of the National palace museum, Taibei. (**b**) Huang Gongwang, The part of *Dwelling in the Fu-chun Mountains* (1350), 33 × 369.9 cm. Courtesy of the National palace museum, Taibei.

Furthermore, in "*Five Sorrows*" (*Da Rocha's Crucifixion*) in "*Song nianzhu guicheng*", the urban background is replaced with a traditional Chinese landscape, and the number of figures is also reduced compared to the original artwork. The "*Pimacun*" technique used for rocks and mountains originates from Chinese traditional landscape painting, where the painter uses a brush to draw long and short lines evenly and slowly, portraying the shadows of rocks and mountains through the natural curvature of the lines. This technique is similar to the one seen in Huang Gongwang's "*Dwelling in the Fu-chun Mountains*" (Figure 5), which had been widely applied to the Chinese paintings by Dong Yuan, Ju Ran, Mi Fu, Wu Zhonggui, and Dong Qichang. Since then, "*Pimacun*" has become the most common painting language in literati paintings, continuing through the late Qing Dynasty, which also became another painting technique in the literati paintings used by the missionaries.

Although, it is unverifiable whether the missionary Jean de Rocha was capable of employing the literati painting techniques to the "*Song nianzhu guicheng*" by himself, as the supervisor of this book, he at least approved of the illustrator's (Gianni Criveller suggests that Jean de Rocha commissioned a Chinese painter, possibly Dong Qichang or one of his students, to create the pictures) literati modifications to the Christian paintings, which proves that he might have recognized the importance of connecting with the literati community for missionary purposes.

To sum up, the analysis of the artistic painting technique in the illustrations of the "*Song nianzhu guicheng*" suggests that they may have used the artistic painting technique of literati painting. This use could be an implicit improvement by the hired literati painter, or it could be a deliberate instruction given to the illustrators. In either case, early missionaries may have recognized the importance of literati in their mission, which made them not only interpret religious doctrines using Confucian thoughts constructed by literati but also use elements of literati painting to alter Christian paintings.

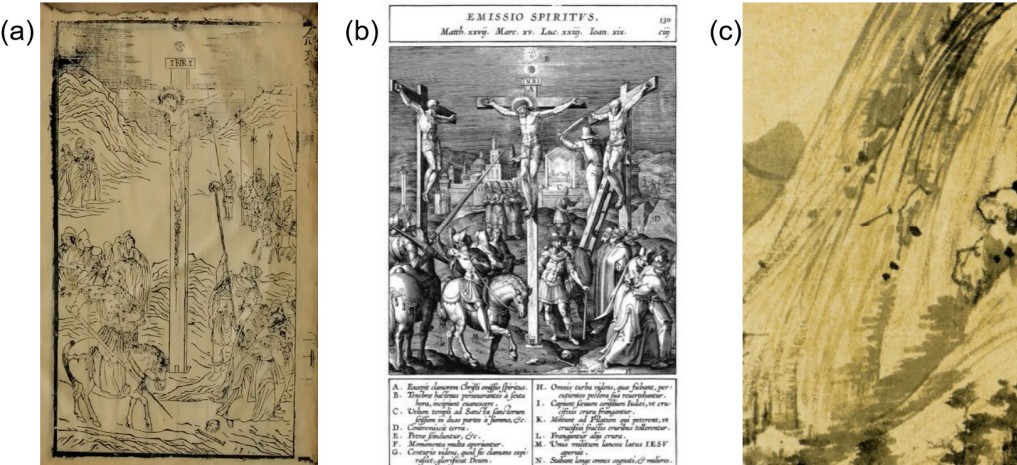

**Figure 5.** Comparison between the "Pima Cun" in "Crucifixion" and Scholar Paintings. (**a**) Crucifixion (drawn from Rocha 1619, p. 56), (**b**) *Crucifixion* (drawn from Nadal 1593, p. 140), (**c**) Huang Gongwang, The part of *Dwelling in the Fu-chun Mountains* (1350), 33 × 369.9 cm. Courtesy of the National palace museum, Taibei.

## 3. The Application of Literati Symbols in Christian Paintings

With the rapid development of the market economy, the economically affluent urban class has widely participated in the artistic activities of the literati community. This behavior has transformed the artistic practices and studies that originally reflected the literati's identity into an aesthetic consciousness collectively advocated by society, known as "literati aesthetics". This aesthetic behavior has led the urban class to actively use certain images or symbols to showcase their shared aesthetic pursuits with the literati. These images can be regarded as "literati symbols". In other words, images such as pine, bamboo, and plum are used in Chinese painting to portray the noble virtues of literati; the landscape screens suggest the mental state of literati; and even symbolized figures like "Dong Qichang" and "Tang Yin" are typical "literati symbols". Artists employ these symbols to place their artworks within the framework of literati aesthetic systems, thereby proclaiming a shared aesthetic taste with the literati.

Similarly, in the illustrations of religious books published by missionaries, we could also observe the use of specific artistic symbols. These symbols often serve as a bridge between the Christian faith and the Chinese cultural context, making the message more accessible and relatable to the literati community.

### 3.1. Landscape Screen

In Chinese paintings, screens with landscape themes were an important element. Wu Hong's research suggests that these landscape screens emerged primarily after the Han Dynasty (202 BC–220), following the advent of individualized Daoist thought that challenged Confucian moral perspectives. By the Song Dynasty (960–1279), these screens had evolved into a theme that complemented the representation of human figures in paintings. For example, the landscape screen in "*The Eighteen Scholars* 十八學士圖" expresses the literati's yearning for natural landscapes, which also makes the landscape screens a common feature of literati studios. However, as the urban middle class gained prominence, landscape screens of the 16th to 17th centuries gradually secularized and became the symbols of "literati" identity, hinting at a vague literati aesthetic. For instance, Du Jin's "*Enjoying Antiquities* 玩古圖" used landscape screens to create an environment for literati appreciation of ancient artifacts (Figure 6). In addition, the 1640 publication of "*The Western Chamber* 西廂記" also incorporated landscape screens multiple times to suggest the literati aesthetic it contained. Hence, landscape screens from the 16th to 17th centuries have become symbolic elements in paintings, which were widely used by the artists to establish connections between their works and the literati community.

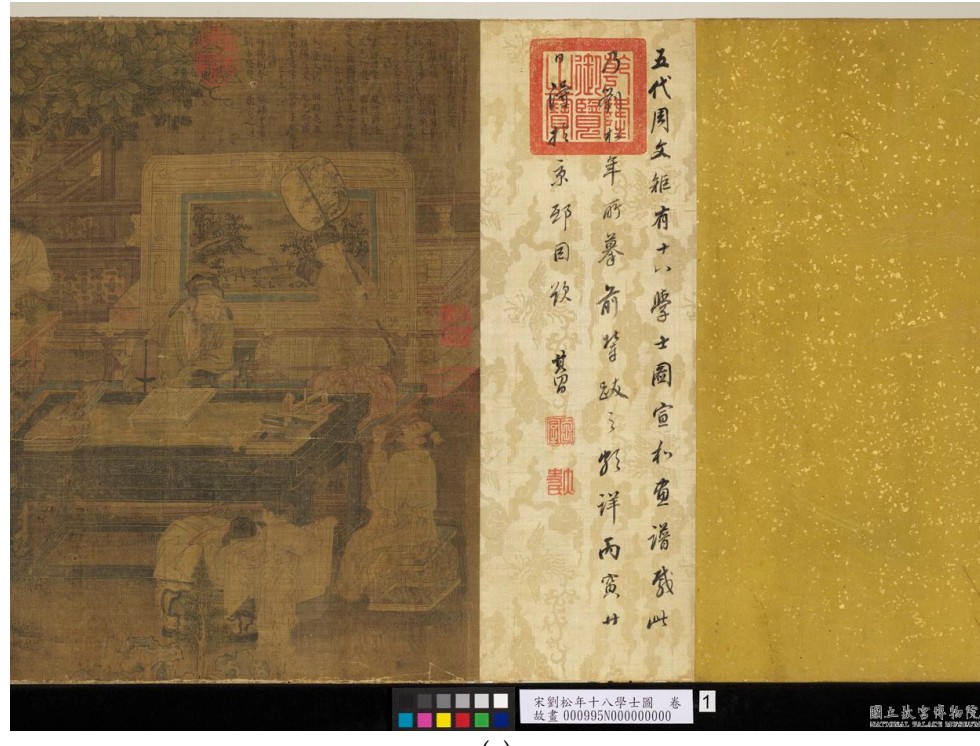

(**a**)

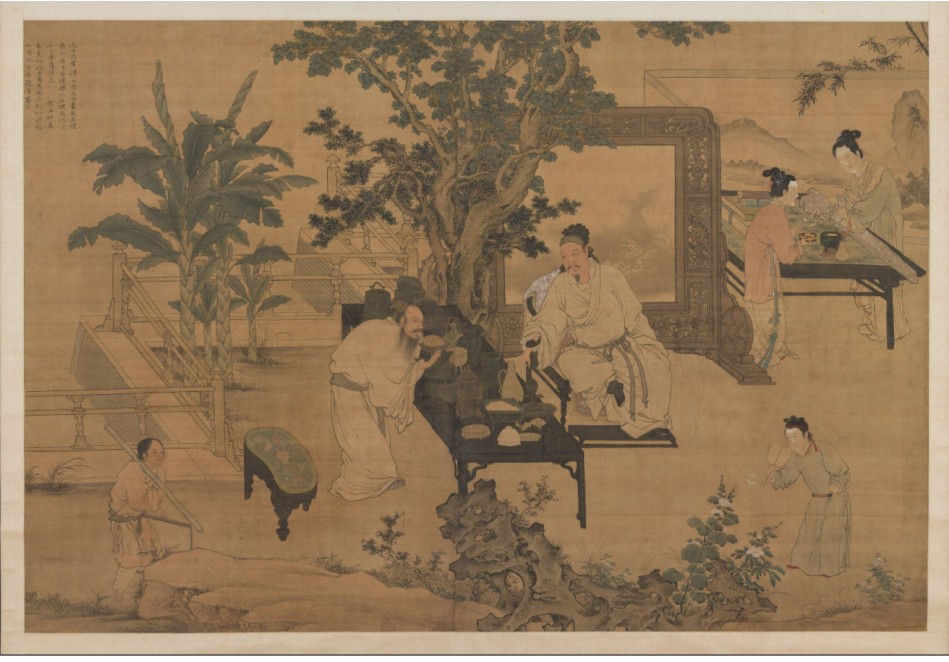

(**b**)

**Figure 6.** Landscape screen in Chinese painting. (**a**) Liu Songnian, The part of *The Eighteen Scholars* (1131–1218), 44.5 × 182.3 cm. Courtesy of the National palace museum, Taibei. (**b**) Du Jin, *Enjoying Antiquities* (1465–1505), 126.1 × 187 cm. Courtesy of the National palace museum, Taibei.

To establish a connection with the literati community, the illustrations in "*Tian zhu jiangsheng chuxiang jingjie*", published by missionary Aleni, also employed this method. Aleni selected 63 illustrations from the "*Evanglicae Historiae Imagines*" and transformed them into 57 Chinese-style woodcut prints for publication. Each illustration in the book has a Chinese title above and Chinese annotations below. In addition to these Chinese characters, we also found that the compiler deliberately changed many compositional elements of the original version and added a considerable amount of common artistic symbols from

Chinese literati painting at the same time. For instance, as shown in Figure 7, there is an illustration in "*Tianzhu Jiangsheng Chuxiang Jingjie*" depicting the "*Maundy Service*" scene from Jesus' miracles. This study is similar to the original composition, presenting three scenes from the story in chronological order. However, compared to the European original, the Chinese woodcut replaces the walls in the background with a landscape screen. In the context of Chinese culture, the landscape screen has special significance, as said by Wu Hong: "The landscape screen became a broad symbol of literature (culture or elegance)" (Wu 1996). Besides that, unlike Rocha, Aleni rather prefers to present the original images in the process of compiling illustrations. This means that, if not authorized by Aleni, the compiler has no right to delete or add elements to the composition. Therefore, the landscape screen in "*Maundy Service*" should be a conscious addition with Aleni's approval, which also becomes evidence for the use of the artistic symbols in the literati painting.

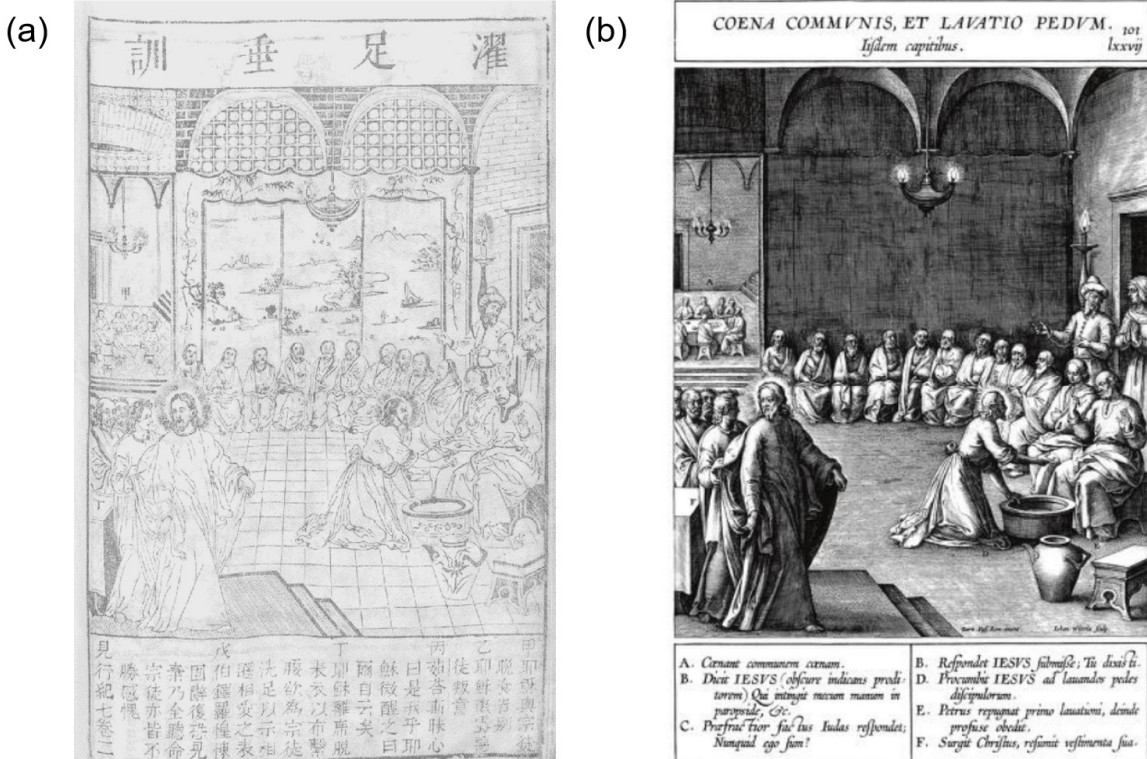

**Figure 7.** The Chinese landscape in the "Maundy Service". (**a**) Christ Washing the Disciples' Feet (drawn from Aleni 1637, p. 52). (**b**) Christ Washing the Disciples' Feet (drawn from Nadal 1593, p. 110).

### 3.2. Dong Qichang

Berthold Laufer (1874–1934) obtained a folded album in Xi'an containing six religious paintings in 1910 (Berthold 1910, pp. 100–18). These paintings were imitated by Chinese artists in the European illustrations, and all six paintings were created by the same person (Noël 1995, pp. 303–14). As shown in Figure 8, one of the paintings in the album directly imitated core figures from the 65th plate of "*Evanglicae Historiae Imagines*". Although the sources for the other paintings in the album are not explicitly identified, it is certain that all six paintings depict scenes from the life of Jesus. One of the paintings in the album bears an inscription and a seal in the lower left corner, with the content of the seal now blurred. However, the inscription is still visible, containing the four characters "*Xuanzai biyi*" (imitate Dong Qichang's painting style). This is the courtesy name of Dong Qichang. Dong Qichang (1555–1636) was a prominent Chinese painter, calligrapher, and art theorist during the late Ming dynasty. Known for his significant contributions to the literati paint-

ing tradition, Dong Qichang played a crucial role in defining and shaping the art scene of his time. The authorship of this set of albums remains unknown, but it is evident that the album is somehow associated with Dong Qichang (Figure 9).

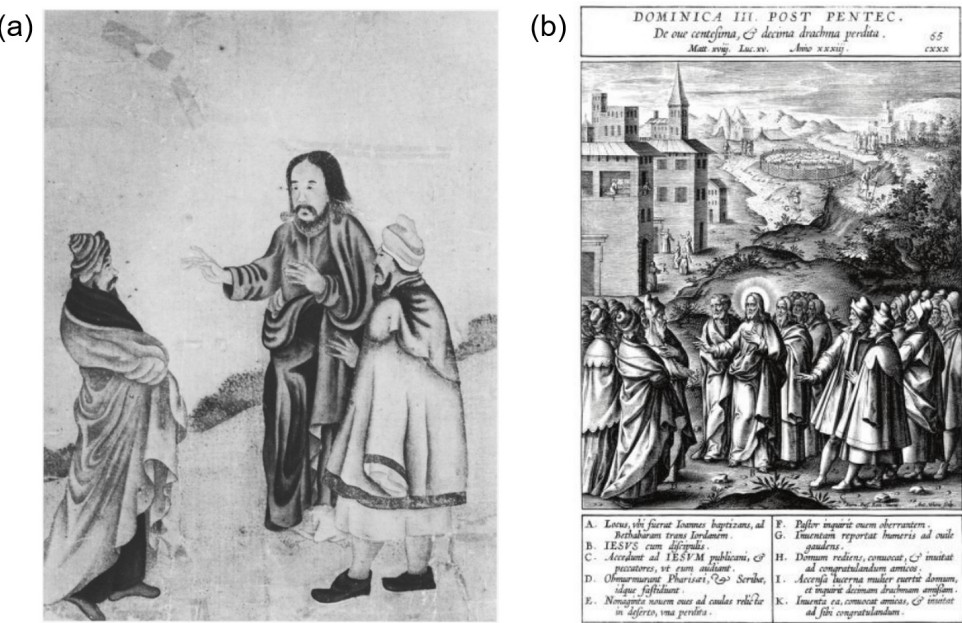

**Figure 8.** A comparison of the engravings in the album and the prints in the Evangelicae Historiae Imagines. (**a**) Anonymous, *Christ on the Road to Emmaus* form one of six leaves "in the style of Xuan Zai" (drawn from Berthold 1910, Plate III) (**b**) *Christ on the Road to Emmaus* (drawn from Nadal 1593, p. 74).

James Cahill's early study of this album asserted that Dong Qichang was influenced by European art through contact with Jesuit missionaries. In addition, Richard Barnhart also supports James Cahill's point and believes that Dong Qichang incorporated European elements into his own creations during his stay in Nanchang in 1597, and the study of the album bearing the inscription "*Xuanzai biyi*" is potential evidence of Dong Qichang's study of European painting. (Barnhart 1997, pp. 7–16). However, Dong Qichang has no reliably transmitted figure paintings, and this album differs from his landscape painting techniques. Thus, to understand "*Xuanzai biyi*," we must consider the literati symbol meaning of Dong Qichang in the unique context of the literati society at that time, not just painting techniques.

Dong Qichang was regarded as an idealized exemplar of the literati community in the late Ming for the following three reasons: first, Dong Qichang's artistic achievements were exceptional with his landscape paintings, which were praised by Zhu Mouyin 朱謀垔 (1584–1628) as "the best in the Ming Dynasty." In addition, the "*Southern and Northern Schools Theory*" proposed by Dong Qichang classified landscape painters after the Tang Dynasty and influenced the aesthetic orientation of artists from the late Ming to the Qing Dynasty profoundly. Second, Dong Qichang's paintings focused on expressing mood and often used elements such as landscapes, peculiar rocks, and trees with delicate painting techniques to convey his understanding and appreciation of natural scenery, which perfectly matched the aesthetic psychology of the literati during that period. After the 17th century, Dong Qichang's artistic style was praised and inherited by literati artists like Wang Shimin 王時敏 (1592–1680), and the subsequent literati also emulated and studied Dong Qichang's artistic style, which further solidified his status within literati society.

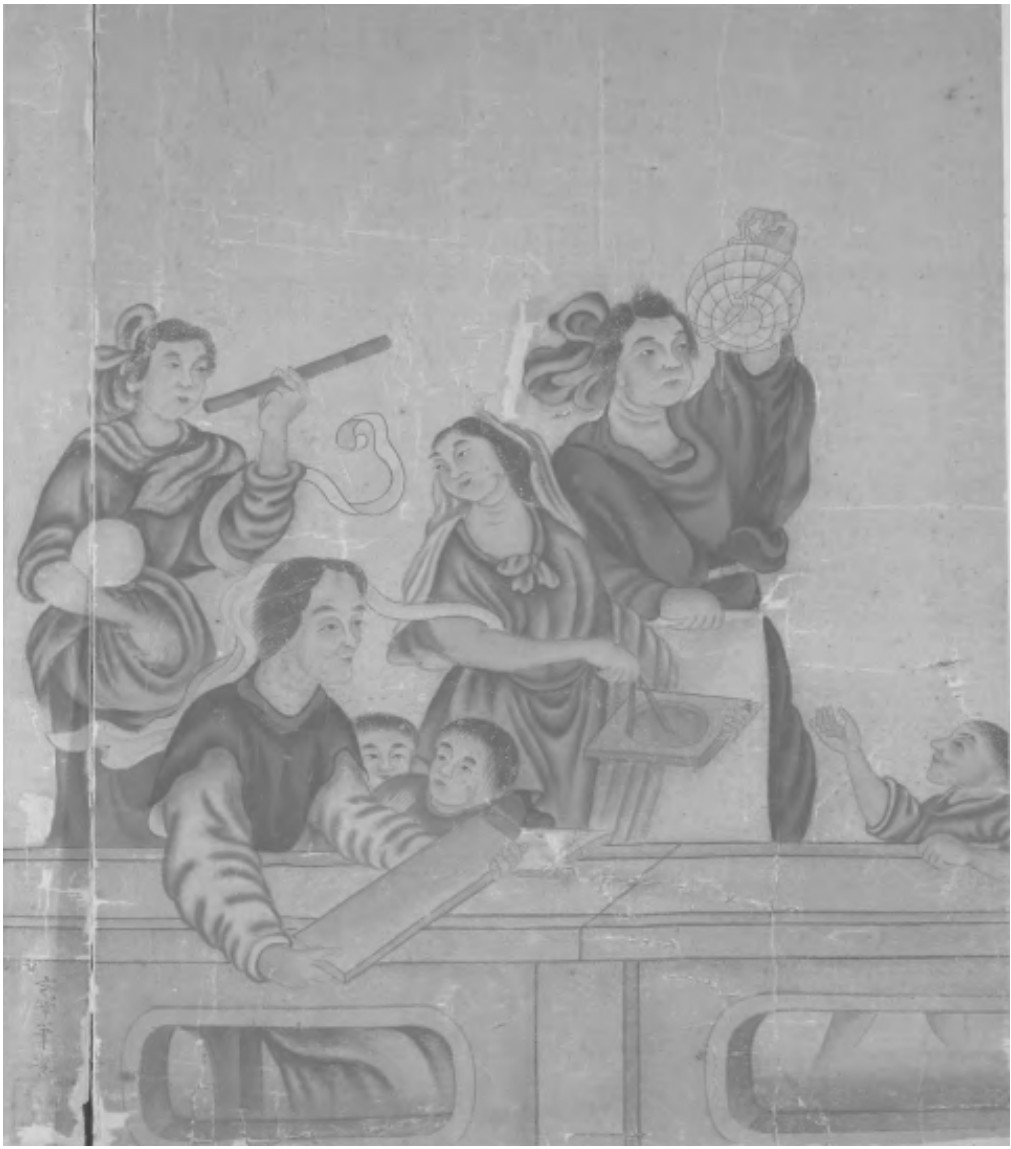

**Figure 9.** Anonymous, Allegory of science and Mechanics form one of six leaves "in the style of Xuanzai" (imitate Dong Qichang's painting style) (drawn from Berthold 1910, Plate VI).

Then, in the 17th century, "*Dong Qichang*" had already become a symbol representing the aesthetic preferences of literati, and the merchant actively purchased artworks bearing the inscription "*Dong Qichang*". As a lower class in the Confucian social order, businessmen sought to establish connections with the higher ranking literati after acquiring wealth, thus enhancing their social status. The merchant's intention to purchase the artworks bearing the inscription "*Dong Qichang*" was actually based on the symbolic significance of Dong Qichang as a literati symbol, rather than the quality or authenticity of the study itself.

In addition, the possibility that the painting was an imitation of Dong Qichang's figure painting techniques was also eliminated, which is because landscape painting was the main type of painting in Dong Qichang's period (Yu 2011, p. 53), and all current research on Dong Qichang indicated that there are no reliable figure paintings of Dong Qichang left behind. Hence, we believe the picture of "*Xuanzai biyi*" in the American Museum of Natural History collection might just present the motivation of missionaries to establish a connection with the literati community by using the *Dong Qichang* as literati symbols for reducing the literati community's resistance to foreign civilizations. Besides that, the inscription of "*Tang Yin*" in the painting titled "*The Madonna Scroll* 中國聖母子像" of the Field Museum of Natural History also plays a similar role, which proves that the applica-

tion of literati symbols has been widely used to establish the connection with the literati community in the missionary process (Figure 10).

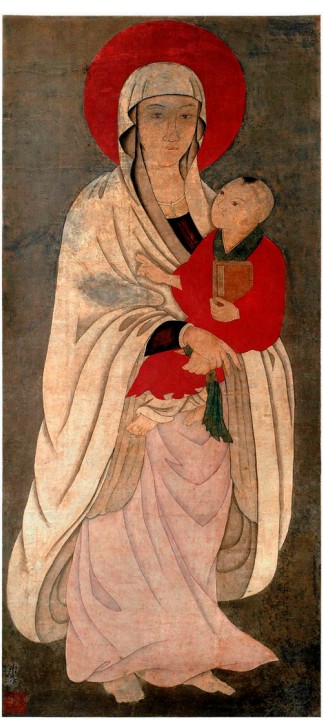

**Figure 10.** Tang Yin, *The Madonna Scroll* (1368–1644), 55 × 120 cm. Courtesy of the Field Museum of Natural History, Chicago.

## 4. Improvement of Results

The transformation of Christian paintings into the literati style by missionaries quickly garnered a response from the literati community. Wu Bin (1550–1643) incorporated Western religious painting's perspective into Chinese landscape painting in his work "*Record of Annual Events and Activities: The Great Exorcism* 歲華紀勝冊:大儺". This new perspective was achieved by using the principle of perspective, where near objects appeared larger than distant ones, especially in architectural elements' portrayals (Figure 11). James Cahill also discussed this perspective shift in "*Distant Mountains: Chinese Painting of The Late*". But the most important thing is that the literalization of religious painting had a significant impact on the literati's hostile attitude towards Catholicism, sparking a trend among them to actively study Western practices. This trend attracted lower level intellectuals without official positions and high-ranking officials, who actively participated in translating, introducing, and promoting Christianity. They also established friendships with missionaries and attended to their daily lives. Xu Guangqi (1562–1633), Yang Tingyun (1557–1627), and Li Zhizao (1565–1630) were among the most outstanding representatives of these literati, collectively known as the "Three Pillars of Chinese Catholicism".

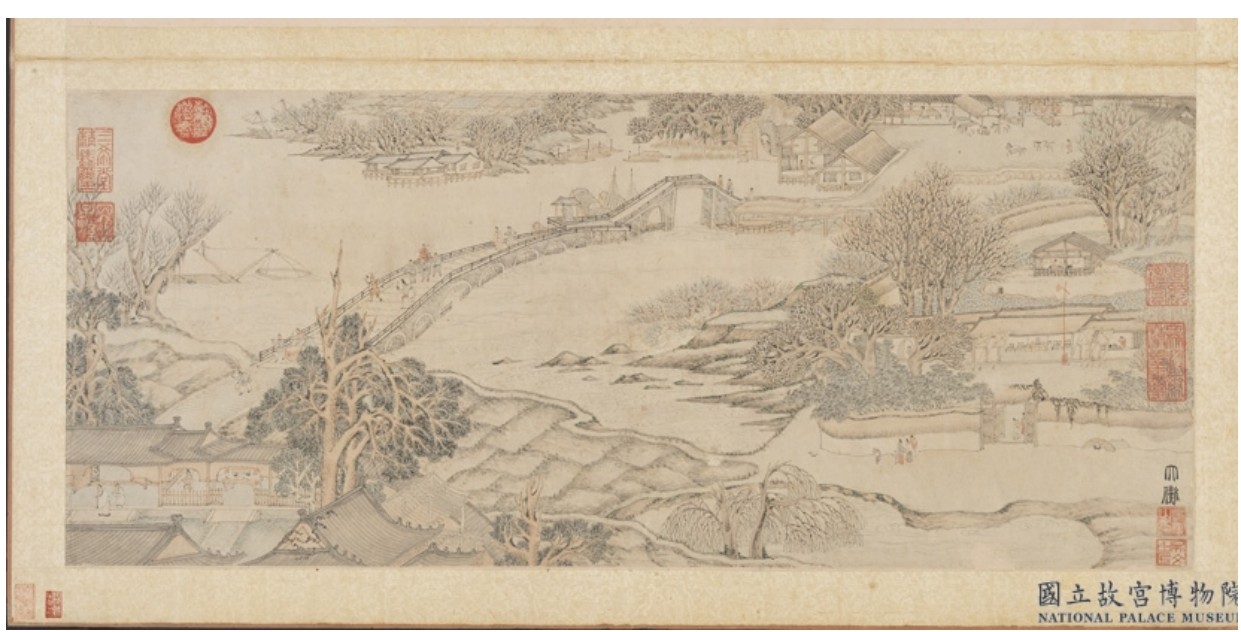

**Figure 11.** The use of perspective. Wu Bin, *Record of Annual Events and Activities: The Great Exorcism* (1550–1643), 29.4 × 69.8 cm. Courtesy of the National palace museum, Taibei.

However, these reforms did not diminish cultural differences between East and West. As Catholicism gained popularity among the common people, the literati community instead became a significant force opposing it. In the "*Nanjing Incident*" of 1616, over 20 foreign missionaries and Chinese converts were arrested. Then, the Fujian government's decree warning against harboring missionaries led to the expulsion of missionaries from various locations in 1637, and 13 churches in Quanzhou were repurposed. In 1639, Xu Changzhi 徐昌治 (1582–1672) compiled an anti-Catholic anthology titled "*Poxie ji* 破邪集 (A collection of papers opposing Catholicism)", which became the most comprehensive anti-Catholic work in the late Ming period. What's more, the anti-Catholic sentiment also extended to religious paintings, and the late Ming literatus Yang Guangxian 楊光先 (1597–1669) believed that the crucified Jesus in the crucifixion was a leader of rebellion and not a law-abiding citizen. This misconception was partly due to early missionaries' efforts to downplay the violent aspects of the "*The Crucifixion of Jesus*" image to cater to literati preferences for non-violence. However, we believe that what missionaries do not know is that the countermove from the literati community is actually because of the missionaries' growing cultural influence on the common people and lower ranking literati, which challenged the literati's cultural leadership. It actually represents the dominant influence of a particular cultural or literary group on shaping societal values, aesthetic standards, and knowledge transmission, which was controlled by the literati community in ancient China. In the 17th century, due to the prevalence of Confucianism in society, literati groups based on the Confucian order continued to play leadership roles and had a vested interest, which made upholding the Confucian social order and moral traditions crucial to their existence. Consequently, the literati community was committed to protecting the privileged position of Confucianism and preventing the influence of Catholicism or other religions from undermining Confucianism's status.

However, the spread of Catholicism in China had a significant impact on the traditional social order. Catholicism and Confucianism have different cultural attributes, which challenge the Confucianism-based social order from different perspectives. For instance, the book "*Jiao you lun* 交友論" written in Chinese by Matteo Ricci faced opposition from some literati before the countermove (Matteo 1948, p. 9). The book emphasizes the importance of friendship and advocates for equal coexistence among friends, regardless of wealth inequality (Shen 2007, pp. 293–94). The literati community criticized this viewpoint



for violating the hierarchical order of Confucian society. Besides that, as the literati community has a vested interest in the Confucian social order, they would not sacrifice their primary position of respecting Catholicism, which has caused a setback in missionary work since the "*Nanjing Incident*" in 1616.

## 5. Conclusions

In the 17th century, the modification methods of the Chinese Christian paintings created by the missionaries were analyzed with the Chinese Catholic studies of "Song nianzhu guicheng" and "Tianzhu Jiangsheng Chuxiang Jingjie". After careful study of the differences between the Chinese Christian painting and the original European version, the analysis results reveal that the missionaries integrated Chinese literati painting elements and symbols, such as "*yudiancun*", "*pimacun*", landscape screens, and the marks of famous literati, into the religious paintings to align with literati aesthetics, thereby hoping to connect with the literati community, achieve their proselytization purposes. Initially, the transformation of Christian paintings into the literati style by missionaries was effective. It caught the attention of the literati community, leading to a surge of interest in "Western learning" in late Ming society. However, the missionaries failed to recognize that the literati community would not abandon the existing Confucian culture and social order to support new ideas of "liberty" and "equality" in Catholicism, which resulted in huge setbacks in their missionary work since the "*Nanjing Incident*" in 1616.

Nonetheless, the Catholic reforms in China were not entirely unsuccessful. While the modifications to Christian paintings did not gain widespread acceptance among the literati, some of the enlightened literati are still willing to learn and accept the advanced thought in the Catholic doctrines. These enlightened literati hoped to explore a new way to solve the societal issues of their period by utilizing the fresh perspectives of European civilization, which represent a significant attempt in the process of Sino-Western cultural exchange.

**Funding:** This research received no external funding.

**Data Availability Statement:** No new data were created or analyzed in this study. Data sharing is not applicable to this article.

**Conflicts of Interest:** The author declares no conflict of interest.

## Notes

[1]    There are two known versions of the record in the book. The first version includes a dialogue between a teacher and students introducing the content of the "Rosary" and how to recite it. Woodcut illustrations and corresponding text explanations are also provided. The second version was printed in Beijing around 1638, lacking the teacher–student dialogue found in the Nanjing edition, but its content is similar to the Nanjing version. However, the Beijing version exhibits stiffer lines and rougher depictions compared to the Nanjing version in terms of woodcut style.

[2]    Both the recitation of the *Rules for Reciting the Rosary* and the *Explanations of the Scripture with Images of the Lord of Heaven Incarnate* are based on the Spanish Jesuit Jerónimo Nadal (1507–1580)'s *Evangelicae Historiae Imagines*.

[3]    *Evanglicae Historiae Imagines* is a collection of religious prints and engravings created by the Flemish artist Joannes David, also known as Jan David or Johannes a Daventria. The study was first published in Antwerp in 1538. The collection consists of a series of visual depictions illustrating various scenes from the New Testament, primarily focusing on the life, teachings, and miracles of Jesus Christ. These images served as a visual aid for conveying biblical narratives to a wider audience, especially those who might not have been literate.

[4]    Chiaroscuro is an artistic technique commonly used in the visual arts, particularly painting, drawing, and photography. It involves the strong contrast between light and dark elements within an artwork, creating a sense of three-dimensional volume and dramatic effects of light and shadow. This technique has been widely used by artists throughout history to achieve depth, contrast, and a heightened sense of realism in their studies.

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
