# Peer review of "Literati Ingredients in the 17th-Century Chinese Christian Paintings"

_religions, doi:10.3390/rel15040383_

Round 1

Reviewer 1 Report

Comments and Suggestions for Authors

Please refer to the attached document!

Comments on the Quality of English Language

Cf. to the attached document!

Author Response

Dear Reviewer,

Thank you for taking the time to review this manuscript. These professional advices you provided is very useful for the further improvement of the manuscript.

Comments 1:In general, the form is good! The structure of the article is good. The illustrations by photos of paintings contribute to the argumentation and coherence of the article. There are a few corrections!

Response 1:According to the attached table, the language issues and spelling errors had been revised carefully and the revised manuscript had been uploaded to the attachment. Thanks for your contribution to this article and these modifications will enhance the quality of the article significantly.

Comments 2:The content is good! The good knowledge of the author on Chinese culture (or at least on Confucianism) in the domain of painting, on the one hand, and the understanding of the Christian aesthetic theology of paintings and icons, on the other, has enriched the article.

Response 2:Thank you for your recognition of the content of this article.

Reviewer 2 Report

Comments and Suggestions for Authors

This article titled "Literati ingredient in the 17th Century Chinese Christian paintings" presents a compelling analysis of how missionaries adapted Christian iconography to align with the aesthetics and cultural symbols of Chinese literati painting during the late Ming Dynasty. The research meticulously compares the modified elements in Chinese Christian paintings with their original European counterparts, highlighting the integration of literati elements such as "Yudiancun" and "Pimacun" strokes, landscape screens, and the influence of renowned literati like Dong Qichang. This study not only deepens our understanding of the cultural exchange between East and West in the 17th century but also sheds light on the missionaries' strategic yet ultimately challenging attempt to engage the literati community in a dialogue on religious and cultural transformation. The paper stands out for its detailed examination of the intersection between art, religion, and cultural diplomacy, making a valuable contribution to the field of art history and intercultural studies.

Author Response

Dear Reviewer,

Thank you very much for taking the time to review this manuscript. These professional advices you provided is very useful for the further improvement of the manuscript.

Comments 1:

This article titled "Literati ingredient in the 17th Century Chinese Christian paintings" presents a compelling analysis of how missionaries adapted Christian iconography to align with the aesthetics and cultural symbols of Chinese literati painting during the late Ming Dynasty. The research meticulously compares the modified elements in Chinese Christian paintings with their original European counterparts, highlighting the integration of literati elements such as "Yudiancun" and "Pimacun" strokes, landscape screens, and the influence of renowned literati like Dong Qichang. This study not only deepens our understanding of the cultural exchange between East and West in the 17th century but also sheds light on the missionaries' strategic yet ultimately challenging attempt to engage the literati community in a dialogue on religious and cultural transformation. The paper stands out for its detailed examination of the intersection between art, religion, and cultural diplomacy, making a valuable contribution to the field of art history and intercultural studies.

Response 1:

Thank you for your encouraging comments on my article and the revised manuscript had been uploaded to the attachment. This comments succinctly encapsulate the essence of my manuscript. Your feedback has not only enhanced the overarching theme of the study within the realm of "cross-cultural art history" but has also inspired me to delve deeper into the intricate interplay between Christianity and the Chinese literary community in future research.
